# Multi-Sensor Laser System for Electric Guitar Pitch-Detection [note 1]

**DOI:** 10.3390/s24082468

**Published:** 2024-04-12

**Authors:** Alessandro Pesatori, Michele Norgia

**Affiliations:** Department of Electronic, Information and Bioengineering, Polytechnic of Milan, Piazza Leonardo da Vinci 32, 20133 Milano, Italy; michele.norgia@polimi.it

**Keywords:** distance measurement, laser measurement, music, position measurement

## Abstract

To attain a direct MIDI output from an electric guitar, we devised and implemented a sophisticated laser sensor system capable of measuring finger positions. This sensor operates on the principle of optical triangulation, employing six lasers and seven position-sensing detectors that are time-multiplexed. The speed and precision of this sensor system meet the necessary criteria for creating an electric guitar with a direct digital output, perfectly satisfying the application’s requirements.

## 1. Introduction

The predominant technique employed in capturing the acoustic wave produced by a guitar typically revolves around the utilization of a pickup. This specialized electronic device functions to transform the resonating sound waves into an electrical signal, essentially acting as a bridge between the analog nature of the guitar’s sound and the digital realm. The conversion facilitated by the pickup is crucial for subsequent processing and amplification. However, it is worth noting that the analog signal generated by the pickup introduces a limitation when aspiring for real-time MIDI output. The inherent time lag associated with note recognition becomes a pertinent issue, with the process often consuming several tens of milliseconds, particularly noticeable in the lower register where notes tend to linger longer. For professional guitarists, operating within a musical context where precision and immediacy are paramount, any delay surpassing 10 milliseconds is deemed unacceptable in their performance environment. In response to this challenge, the laser pitch-detection system (LPD) emerges as a noteworthy alternative [1], presenting itself as a potential solution in scenarios where the standard pickup technology falls short. The LPD system operates on a different principle, relying on laser technology to detect pitch and note information with remarkable speed and accuracy. Unlike traditional pickups, the laser pitch-detection system bypasses the analog-to-digital conversion process, minimizing the latency associated with note recognition.

In essence, the LPD system stands out as a viable choice for professional guitarists who demand real-time responsiveness and accuracy in their musical endeavors. By circumventing the limitations of conventional pickups, especially in situations where milliseconds matter, the laser pitch-detection system offers a promising avenue for enhancing the performance capabilities of guitarists in a variety of musical settings.

The primary aim of this innovative project is to design and implement a sophisticated system that can conduct real-time measurements of finger positions during guitar playing, subsequently translating this data into musical MIDI output in response to the plucking of strings, as detected by a specialized pickup. To comprehend the operational intricacies, refer to Figure 1, which visually outlines the conceptual framework: a set of six laser beams aligned in parallel along the strings, extending towards the nut of the guitar.

When a guitar is played, the player’s fingers come into play by shortening the distance between the string and the fingerboard through the act of pressing. This crucial moment is where the magic happens—the finger interrupts the laser beam, causing the light to be reflected back. In the context of this study, we introduce an advanced measurement system built upon the principle of laser triangulation, enhanced by the presence of multiple photodetectors strategically positioned to identify and record all potential finger positions. The goal is to harness this information and seamlessly transform it into a direct and dynamic MIDI output. The utilization of laser triangulation marks a pivotal departure from traditional methods, offering a precise and comprehensive approach to capturing the intricacies of finger positioning during guitar performance. By integrating multiple photodetectors into the system, we aim to achieve a granular level of detail, ensuring that every conceivable finger position is accurately identified and accounted for. This meticulous approach is paramount in generating a MIDI output that faithfully mirrors the nuances and subtleties of the guitarist’s playing technique.

In summary, this project stands as a pioneering effort to revolutionize the intersection of technology and music by developing a cutting-edge system that not only facilitates the real-time measurement of finger positions but also transforms this data into a seamless and expressive MIDI output. Through the integration of laser triangulation and sophisticated photodetection mechanisms, our approach promises to usher in a new era of precision and musical fidelity in the realm of guitar performance.

## 2. Laser Sensing Technique

The selection of the optical technique for distance measurement must consider the demanding performance requirements specific to the application, which are quite stringent:Laser safety class II: optical power lower than 1 mW.Response time: lower than 2–3 ms.Absolute distance range: about 1 m.Distance resolution: about 1 mm.Absolute distance accuracy: about 1%.No crosstalk between adjacent laser sensors.Infrared laser emission.Low cost of the single channel.Possibility of battery-operation—low power consumption.

In addressing the specified requirements, one potential avenue worth exploring involves the utilization of a time-of-flight measurement system [2,3,4,5], employing either pulsed or continuous-wave lasers. However, this approach, commonly employed for long-distance measurements, proves unsuitable in the present context due to its inherent limitations in resolution and high associated costs. Another option lies in coherent measurement systems [6], including those leveraging synthetic frequencies [7]. Despite their intricacy, these techniques fall short of meeting the necessary criteria for speed and resolution. Alternatively, a distance meter based on the self-mixing effect [8,9], elaborated upon in the literature [10,11], emerges as a viable choice. While the initial implementation of this system demonstrated effective operation, challenges arose when working with fingers lacking cooperative targets. A primary obstacle in this scenario was ensuring laser safety, given that the system operated in real-time with an optical power exceeding 5 mW of continuous power. A third and final solution, thoroughly investigated in this study, revolves around the utilization of an optical triangulator [12,13,14,15,16]. This approach strikes a delicate balance between complexity, cost, and performance. Optical triangulation-based distance measurement systems, known for their versatility, find applications across diverse industrial and scientific contexts, spanning both distance [2,17] and vibration [18,19,20] measurements. Figure 2 intricately illustrates the operational principle of the optical triangulator-based instrument: the laser spot impacts the position-sensing detector (PSD) [2,3], and the position of impact is meticulously determined by the absolute distance to the target, establishing an inversely related correlation between these two pivotal parameters. This multifaceted and adaptable system offers a promising solution, aligning with the study’s objectives while considering the intricate balance between technical intricacies, economic feasibility, and optimal performance.

The equation governing the instrument is found by the following geometrical consideration:(1)$$L=\frac{D}{x}f,$$
where *L* is the target distance, *D* is the triangulator baseline (distance between laser and lens), *f* is the lens focal length, and *x* is the spot position in the PSD. By deriving the equation with respect to *x*, we find the following:(2)$$\frac{∆L}{L}=-\frac{∆x}{x},$$

As a result, the resolution of the distance measurement, denoted as Δ*L*, is directly proportional to the resolution of the position-sensing detector (PSD), indicated as Δ*x*.

To mitigate disruptions caused by ambient light, a pulsed laser source is employed, and the receiver is equipped with a high-pass filter, typically set above 100 Hz (the typical frequency of artificial light sources). Furthermore, for the LPD system to accurately measure the positions of fingers across the guitar’s six strings, it necessitates six laser sources. These lasers are multiplexed, ensuring that only one laser diode is active at any given time. This approach guarantees that only the reflections from a specific laser, assigned to a particular string, are utilized for the calculations. When the multiplexing frequency exceeds 6 kHz, the recognition delay remains under 1 ms, regardless of the string’s frequency. This means it performs equally well for bass instruments. It is worth noting that this latency applies to pitch recognition without the requirement to pluck the string.

Our decision was to implement a pulsed laser system with a repetition frequency of 6 kHz and a duty cycle of 10%, resulting in pulses lasting approximately 15 µs.

## 3. System Design

To develop the geometry of the system, we calculated the theoretical resolution of the sensor, as a function of the back-diffused light and of the triangulator baseline (*D* in Figure 2). The noise current of a PSD depends on the inter-electrode resistance [2,21]. For the noise calculation, we focus on a commercial PSD: model OD3.5-x from Silicon Sensors, with an inter-electrode resistance *R* = 50 kΩ. For this value, we can estimate a PSD noise current *i*_noise_ = 0.6 pA/√Hz.

We consider a pulse duration of about 10 µs; therefore, we can suppose a noise-equivalent bandwidth *B* = 100 kHz. The noise current integrated on *B* is equal to the following:(3)$$I_{noise}=i_{noise}\cdot \sqrt{B}≅0.19\;\mathrm{nA},$$

To find the signal current, we need to estimate the optical power impinging on the PSD. We assume that the finger is an optical diffuser, with emissivity ε = 0.7 and a Lambertian distribution of the scattered light. The fraction *P*_sensor_ of the emitted light *P*_laser_ collected by the lens is calculated as a fraction of the Lambertian emission on half sphere (π*L*^2^):(4)$$P_{sensor}=\frac{S_{lens}}{\pi L^{2}}\cdot \varepsilon \cdot P_{laser},$$

Considering a lens surface *S*_lens_ = 0.5 cm^2^ (circular lens with diameter ~7 mm), at the maximum distance *L*_max_ = 65 cm, we found *P*_sensor_ ≅ 26 × 10^−6^⋅*P*_laser_. With a peak power of the pulsed laser source equal to *P*_laser_ = 5 mW, we calculate an optical power on the PSD equal to *P*_sensor_ = 132 nW.

The considered PSD exhibits a responsivity equal to 0.59 A/W at a wavelength of 850 nm; therefore, the global signal current is *I* = 78 nA.

The spot position x on the PSD is given by [2]:(5)$$x=\frac{I_{1}-I_{2}}{I_{1}+I_{2}}=f\left(I_{1},I_{2}\right),$$
where *I*_1_, *I*_2_ are the two currents coming out from the PSD.

By calculating the uncertainty of the spot position on the PSD as a function of the noise and signal currents, we obtain a noise-limited resolution between 2 µm and 3 µm.

This noise calculation enables the design of the geometry for the whole system. By assuming a worst-case PSD resolution of 3 μm (limited by noise) and a PSD length of 3.5 mm, we simulated the range of the triangulator as a function of the baseline *D*, with the lens focal length *f* as a parameter. In Figure 3, we report the working dynamics for the final choice *f* = 15 mm. The measurable range is calculated by the Signal-to-Noise ratio and the geometry of the triangulator, to reach a resolution of 1 mm (blue line, down), 2 mm (green line, center), or 3 mm (black line, up). The focal length was decided as a tradeoff between range, resolution, and encumbrance.

The minimum measurable distance is limited by the PSD length; the maximum distance is limited by the desired accuracy: it can be calculated from Equations (1) and (2). To calculate the maximum distance (*L*_max_), the PSD length (*x*_PSD_) needs to be taken into account, which limits the minimum measurable distance. You have to substitute these values to *L* and *x* in Equations (1) and (2). In particular, in Equation (2), you only have to consider the absolute value $\frac{∆L}{L_{max}}=\left|\frac{∆x}{x_{PSD}}\right|$. Substituting $x_{PSD}=∆x\cdot \frac{L_{max}}{∆L}$ in $L_{max}=\frac{D}{x_{PSD}}f$, you obtain the following:(6)$$L_{MAX}=\sqrt{\frac{∆L}{∆x}\cdot D\cdot f}$$

The guitar system needs measurements in the range 20–65 cm; therefore, we have to design the number and position of the PSDs in order to keep a good resolution in that distance range. The final geometry, realized for the best performances and capability of measuring each finger position, is described in Figure 4.

This final system is constituted by seven detectors, with seven lenses (10 mm diameter, focal length 15 mm), alternated to the six lasers. We decided to implement more receivers; in this way it is possible to detect the finger position, even in the case of a partial shadow of another finger. Figure 5 shows the 3D design of the mechanical mounting and Figure 6 shows the electronic board developed with all the optics, the lasers, and the PSD.

## 4. System Development

The realization and test of the first prototype was made with a PC and a Data Acquisition Board. The final system employs an FPGA for controlling the timing of the laser pulses, and the sampling of the analog signals from the seven PSDs. The pulse duration is 40 µs, acquired after using an analog high-pass filter (cutoff frequency at 4 kHz). The high-pass filter is at a quite high frequency, to better cancel the ambient light, and to avoid crosstalk between adjacent pulses: the transitory character of each pulse vanishes when the next pulse occurs. All the next calculations are made by a DSP. For each laser pulses and each PSD, it calculates the sum and the difference of the two currents *I*_1_ and *I*_2_, and the laser spot position *x* on the PSD.

In order to also consider the non-linearity of the system, we decided to implement a characterization curve of the following kind:(7)$$L=\frac{1}{c_{1}+c_{2}x+c_{3}x^{2}+c_{4}x^{3}}$$

The values of these four constants are determined through a least-squares regression technique applied to nine calibration points, spaced at intervals of two guitar frets. This calibration process is automated and integrated into the DSP (Digital Signal Processor). It prompts the user to place a finger as a barre, starting from a specific guitar fret and moving toward the neck. The DSP then records and averages the signals from the seven position-sensing detectors (PSDs) for all six lasers, corresponding to each finger position, and subsequently calculates the regression curves.

Figure 7, Figure 8 and Figure 9 depict three examples of calibration curves obtained for the second laser using PSDs numbered 1, 3, and 7, respectively. The dots represent the calibration points, while the line represents the regression curve, where the Out PSD is equal to (*I*_1_ − *I*_2_)/(*I*_1_ + *I*_2_), and *I*_1_, *I*_2_ are the photocurrents coming from the single one-dimensional PSD inside the electronic guitar. In the case of the outermost PSD, number 7, the closest distances fall outside the acceptable range. Consequently, the final algorithm must account for which receivers provide accurate measurements. The solution to this issue is elaborated upon in the following section.

## 5. Time Pulse Design and Digital Acquisition

To make the distance measurement, an analog electronic board was designed to generate a train of pulses on every laser used to measure the distance of the fingers on the guitar bridge in “real-time”. The designed train of pulses has a duty cycle of 3.3%, with a period lasting 1.2 ms and pulse of 40 μs, as shown in Figure 10. The average power released was 500 μW with a peak of 15 mW, respecting safety requirements. In order to not confuse the system that measures the distance by reading the series of the PSDs, the lasers were switched on sequentially after each 200 μs. In this way, every 1.2 ms was a complete cycle of measurement for the six lasers employed. For this application, the coherence, monomodality, and the wavelength of the laser are not crucial, because there are no specific requirements.

The signal received on the PSDs was analog-filtered with a high-pass filter at 4 kHz, as shown in Figure 11. This eliminates any offset, disappearing before the beginning of a new pulse.

Then, the fourteen signals (two for every PSD, because you have to detect its two current anodes to measure the position of the reflection) are acquired by two A/D converters, multiplexed over seven channels. Each complete single acquisition (i.e., *C*_2_ in Figure 12) requires 20 μs in total: 3 μs for reaching the regime value and 17 μs for sampling seven channels. For each pulse, four samples are acquired. Initially and at the end, the pulses were measured, *S_C_*_1,n_ and *S_C_*_4,n_; the measurements are used to remove the eventual offset before and after the pulse. Then, the pulses were measured again, *S_C_*_2,n_ and *S_C_*_3,n_, from which *S_C_*_1,n_ and *S_C_*_4,n_ are subtracted, respectively. The pulse amplitude *S*_n_ is measured as follows:(8)$$S_{CL,n}=\frac{\left(S_{C2,n}-S_{C1,n}\right)+\left(S_{C3,n}-S_{C4,n}\right)}{2}$$

In this way, all the errors related to offset that can lead to a reduction in absolute accuracy will be significantly reduced at the cost of a simple additional elaboration.

## 6. Elaboration Algorithm and Uncertainty Estimation

We have developed an algorithm for estimating finger positions based on the information generated by different PSDs (position-sensitive detectors). The most accurate measurements can be achieved by calculating a weighted average of the distances estimated by each PSD [22], where the weights are inversely proportional to the square of the measurement uncertainty. Only meaningful measurements should contribute to this average; PSDs without a signal (e.g., when obscured by a finger) should not be included.

The uncertainty in spot position *u*(*x*) over the PSD depends on the signal’s amplitude in relation to the noise and disturbances level [14]. The uncertainty in the current *I* can be expressed as follows: $u\left(I\right)=\frac{I_{\mathrm{noise}}}{I}$.

Among the various sources of noise, we can neglect the contribution of shot noise since the current signals in the PSD are consistently very small. In this scenario, the current noise *I*_noise_ remains independent of the signal amplitude and can be considered a constant term. It is primarily determined by factors such as noise power resulting from the interelectrode PSD resistance, dark current of the sensor, and disturbances.

We can then write the following:(9)$$u\left(x\right)=\frac{c_{1}}{P_{\mathrm{signal}}}$$
where *c*_1_ is just a constant term depending on the noise power.

Finally, since the signal strength *P*_signal_ depends directly on the sum signal *S*_sum_ obtained by the sum of the currents *I*_1_ and *I*_2_ of the PSD (i.e., the total value of the photogenerated current), we obtain the following:(10)$$u\left(x\right)=\frac{c}{S_{\mathrm{sum}}}$$
where *c* is a constant term, considering all the noise and disturbance contributions. It can be determined experimentally.

Considering (1) and (2), we obtain the estimated uncertainty of a single distance measurement *u*(*L*_i_):(11)$$u\left(L_{i}\right)=c\frac{L_{i}^{2}}{D_{i}\cdot f\cdot S_{i\;\mathrm{sum}}}$$

We can formulate the algorithm of distance calculation as the weighted average of the seven distance measurements, *L*_i_, made by the seven PSDs. The weights are inversely proportional to the square of the uncertainty measurement $u{\left(L_{i}\right)}^{2}$ [15]:(12)$$\bar{L}=\frac{\sum_{i=1}^{7}\left(\frac{L_{i}}{u_{i}^{2}\left(L\right)}\right)}{\sum_{i=1}^{7}\left(\frac{1}{u_{i}^{2}\left(L\right)}\right)}=\frac{\sum_{i=1}^{7}\left(L_{i}{\left(\frac{D_{i}\cdot f\cdot S_{i\;\mathrm{sum}}}{L_{i}^{2}}\right)}^{2}\right)}{\sum_{i=1}^{7}{\left(\frac{D_{i}\cdot f\cdot S_{i\;\mathrm{sum}}}{L_{i}^{2}}\right)}^{2}}$$
where *S_i_*
_sum_ is the sum signal of PSD number *i*, and *D*_i_ is the distance between the active laser and the PSD number *i*.

To simplify the algorithm, it is possible to substitute the single estimated values *L_i_*, in the uncertainties, with the exact distance *L*. Its exact value is unknown, but this factor is simplified in (12), so we can use the following simple equation:(13)$$\bar{L}=\frac{\sum_{i=1}^{7}\left(L{\left(D_{i}\cdot S_{i\;\mathrm{sum}}\right)}^{2}\right)}{\sum_{i=1}^{7}{\left(D_{i}\cdot S_{i\;\mathrm{sum}}\right)}^{2}}\;$$

This approximation is also theoretically consistent: the uncertainty of each distance measurement depends on the geometry; therefore, it depends on the exact target distance *L*, not on the estimated distance *L_i_*. The values are included in the average (13) only if *S_i_*
_sum_ is over a threshold; doing so prevents inserting the noise, added by an obscured PSD, in the average.

The complete algorithm only applies (12) to the PSDs that cover the global distance range, with respect to the used laser. For example, the seventh PSD is not used for the initial estimation of a finger on the second cord (see Figure 9). We created a table indicating the “always correct” PSDs for each laser. As a first estimation, the DSP only calculates (12) for the “always correct” PSDs; then, if the distance is higher than about 30 cm, the DSP performs a recalculation using the signals of all available PSDs. In this way we use the information of all the PSDs for far distances, where the resolution of the triangulator is worse (see Figure 3).

With the implemented algorithm, we realized a resolution of about 1 mm with an accuracy of 2–3 mm, in a response time lower than 2 ms. The accuracy achieved is exactly what it is needed to distinguish without any doubt the position of a finger on the guitar and assign it to the corresponding fret and pitch. Figure 13 shows a photo of the realized prototype mounted on an electric guitar.

The global system exhibits very good performances without strings, but there are still some improvements to make when the lasers are shadowed or make multiple reflections hitting the strings. These reflections sometimes can induce an incorrect distance measurement. A complex algorithm that considers this effect and corrects the results is now being studied.

## 7. Conclusions

In the realm of developing a real-time MIDI system for electric guitars, the accurate measurement of finger positions stands as a pivotal aspect. This comprehensive study delves into the intricate design and prototyping of a cutting-edge multi-laser system specifically tailored for guitar detection. The implementation of an advanced real-time multi-sensor algorithm, a cornerstone in achieving the necessary performance criteria, is meticulously detailed. The resulting prototype is a testament to technological innovation, attaining good performances with an accuracy of approximately 2 mm over a range of about 1 m, which is enough to assign the corresponding fret correctly. Equally noteworthy is its remarkable response time, clocking in at less than 2 milliseconds, aligning seamlessly with the stringent requirements imposed by musical applications. An additional feature enhancing its usability and safety is the utilization of a pulse of an infrared laser (980 nm) as the optical source, ensuring that the system emits an optical power that does not exceed 0.5 mW, thereby falling under safety class I. This classification instills confidence in users, as the system can be utilized without any apprehension about potential hazards. Furthermore, the low power consumption opens up the possibility of operating the system without the need for cables, allowing for a wireless experience by relying on a battery to supply power. Future developments of this Multi-Sensor Laser System include the possibility to add some additional MIDI parameters, like velocity and modulation, to enhance the system’s ability to capture the intricacies of guitar performance.

Crucially, the versatility of this system extends beyond the realm of electric guitars. It demonstrates adaptability to various stringed instruments equipped with a fingerboard, irrespective of the presence of frets. Moreover, it accommodates diverse playing styles, whether involving a pick or a bow. This expansive applicability underscores the system’s potential to revolutionize the landscape of stringed instrument technology, catering to the needs of musicians across different genres and playing preferences. The meticulous attention to safety, precision, and adaptability positions this multi-laser system as a promising innovation in the ever-evolving intersection of music and technology.

## Figures and Tables

**Figure 1 sensors-24-02468-f001:**
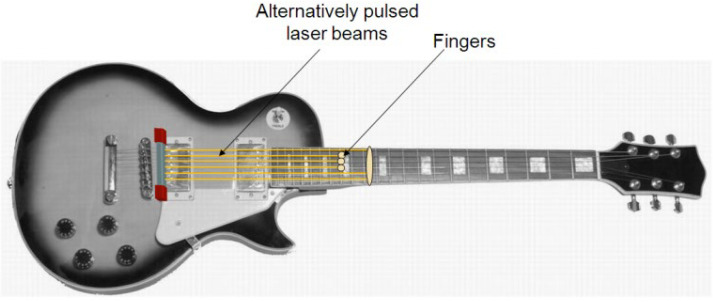
Explanation of the laser pitch-detection system.

**Figure 2 sensors-24-02468-f002:**
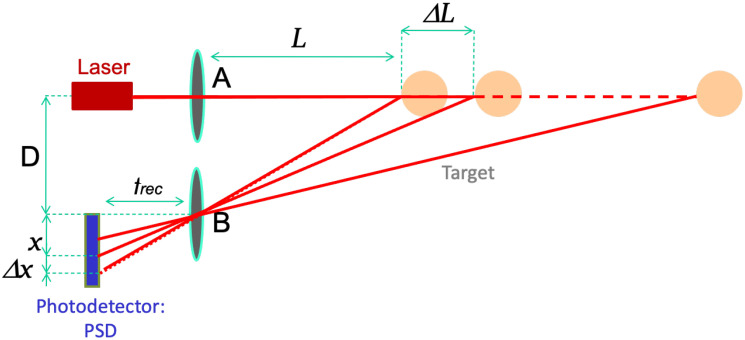
Description of triangulation principle.

**Figure 3 sensors-24-02468-f003:**
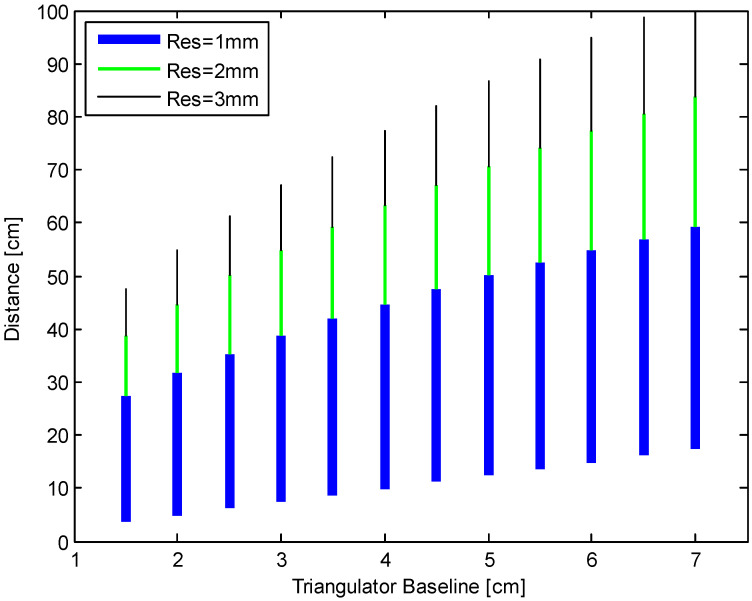
Working ranges of the instrument, as a function of the desired distance resolution and of the triangulator baseline (focal length *f* = 1.5 cm).

**Figure 4 sensors-24-02468-f004:**
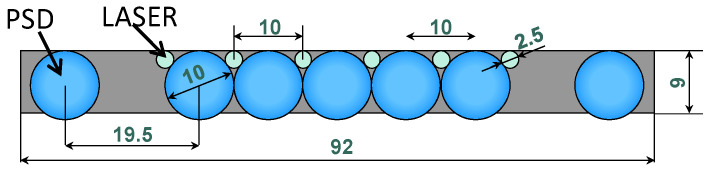
Design of the multiple laser-receiver system: seven lenses for PSD and six lenses for laser collimation. Dimensions are in mm.

**Figure 5 sensors-24-02468-f005:**
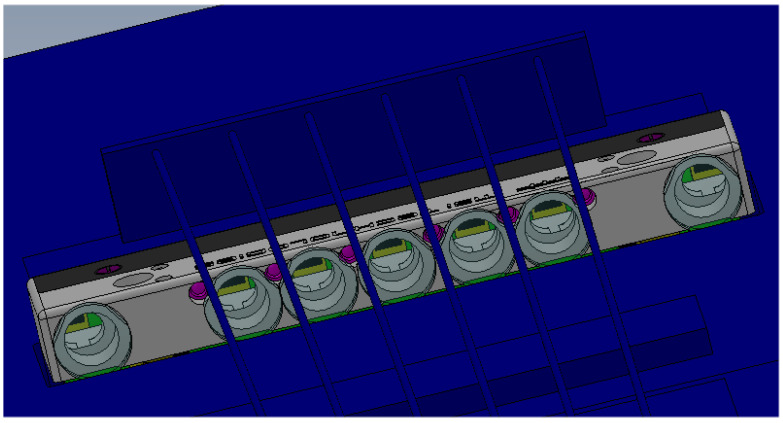
Design of the mechanical mounting for seven PSDs and six lasers.

**Figure 6 sensors-24-02468-f006:**
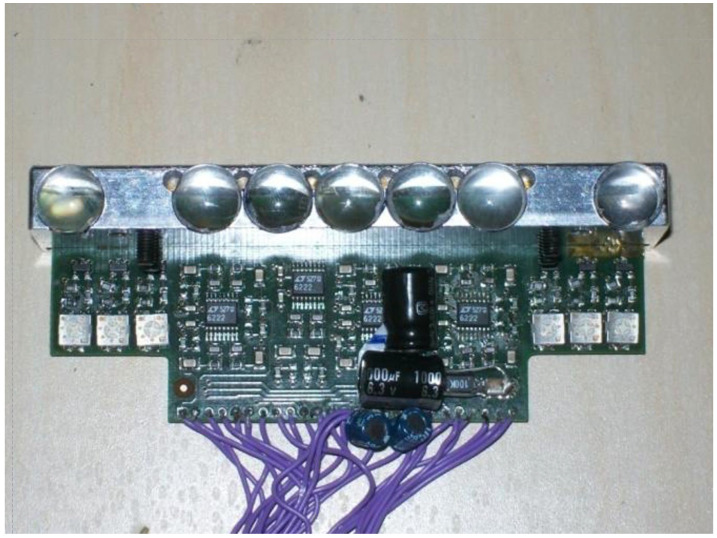
Electronic board developed by mounting seven PSDs, the lasers, and the PSD optics.

**Figure 7 sensors-24-02468-f007:**
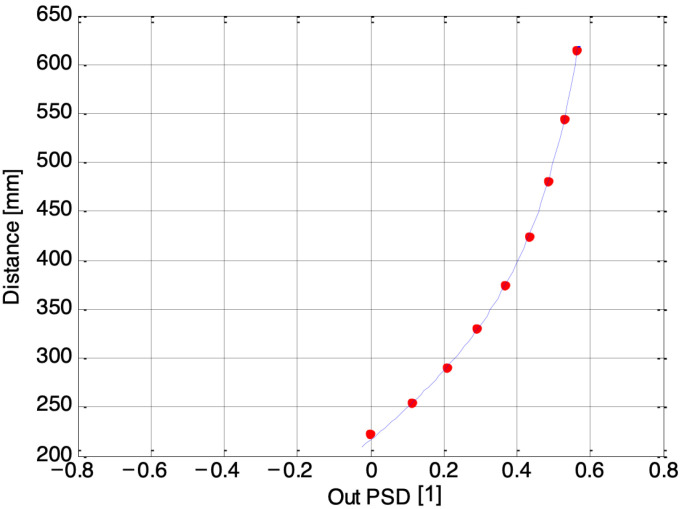
Calibration curve for the second laser, acquired by the first PSD.

**Figure 8 sensors-24-02468-f008:**
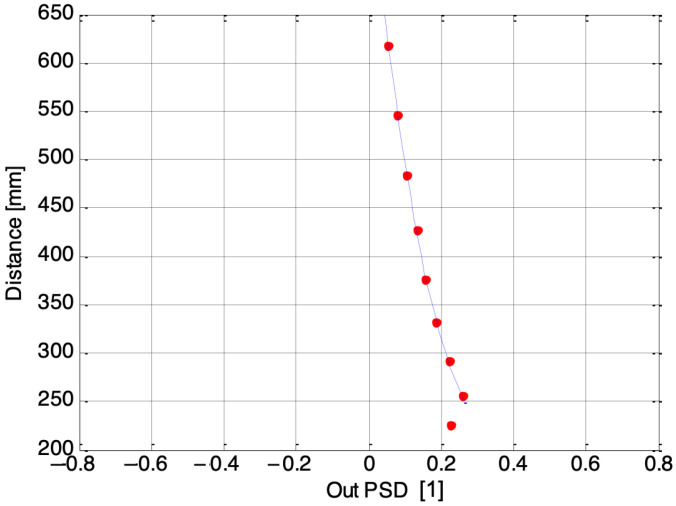
Calibration curve for the second laser, acquired by the third PSD.

**Figure 9 sensors-24-02468-f009:**
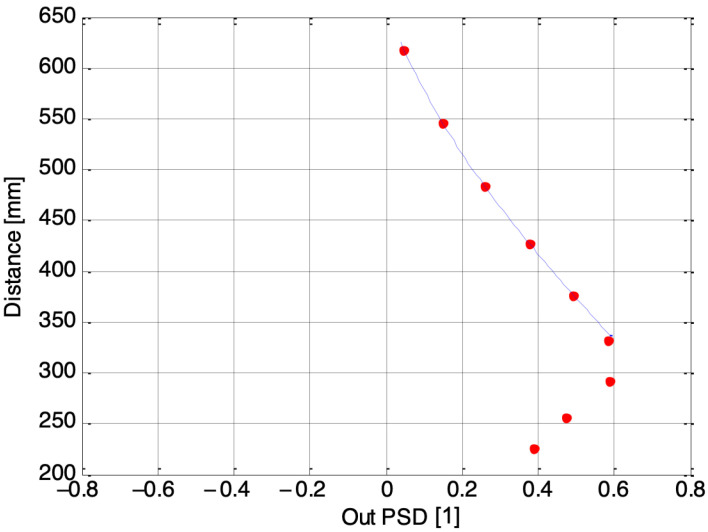
Calibration curve for the second laser, acquired by the seventh PSD.

**Figure 10 sensors-24-02468-f010:**
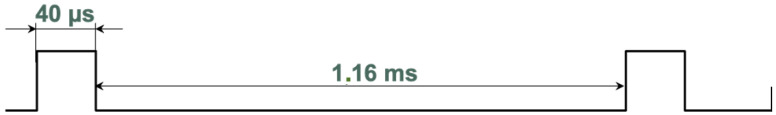
Laser pulse timing characteristic.

**Figure 11 sensors-24-02468-f011:**
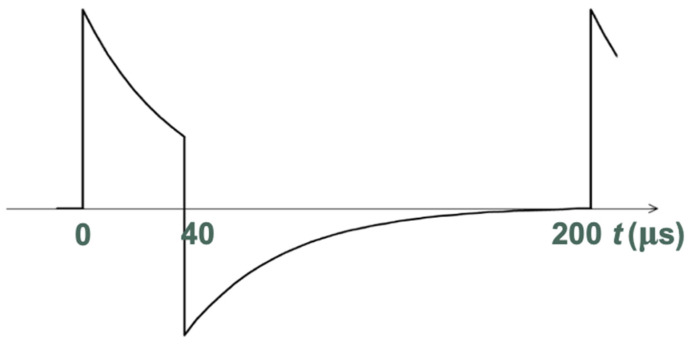
Laser pulse characteristic after the analog filtering.

**Figure 12 sensors-24-02468-f012:**
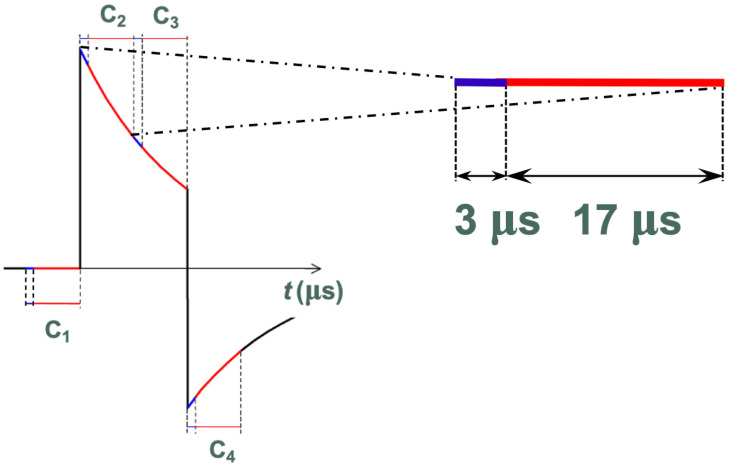
Acquired laser pulse. Every acquisition took 20 μs: 3 μs for reaching the regime value (blue line) and 17 μs for sampling seven channels (red line).

**Figure 13 sensors-24-02468-f013:**
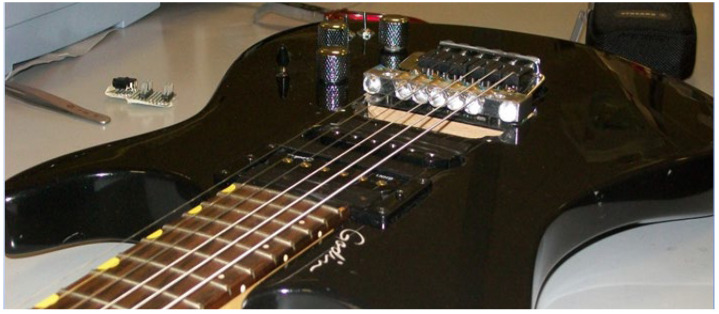
Photo of the realized prototype mounted on an electric guitar.

## Data Availability

Data are contained within the article.

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
