# Peer review of "Multi-Sensor Laser System for Electric Guitar Pitch-Detection†"

_sensors, 2024, doi:10.3390/s24082468_

Round 1

Reviewer 1 Report

Comments and Suggestions for Authors

This paper proposes a laser triangulation system to detect finger positions of guitarist. It is an interesting work. However, I have two concerns:

1. Laser triangulation systems are widely used in many fields and what is your novelty in this paper? Please clarify.

2. I found no results on positioning accuracy. Is the accuracy important? Does it work if we only detect fingers other than getting the accurate positions?

Comments on the Quality of English Language

None

Reviewer 2 Report

Comments and Suggestions for Authors

see the attached

Comments on the Quality of English Language

none

Author Response

Thank your for your review. Please see the attachment.

Round 2

Reviewer 1 Report

Comments and Suggestions for Authors

All my concerns are solved and I think this paper can be accepted for publication.

Comments on the Quality of English Language

None